ecology, genomics

coral bleaching, genotype-by-environment interactions, coral restoration

**Author for correspondence:**
Crawford Drury
e-mail: crawford.drury@gmail.com

†Present address: Hawai'i Institute of Marine Biology, University of Hawaii at Mānoa, Kāne'ohe HI, 96744, USA.

# Genotype by environment interactions in coral bleaching

Crawford Drury† and Diego Lirman

Department of Marine Biology and Ecology, Rosenstiel School of Marine and Atmospheric Science, University of Miami, Miami, FL 33149, USA

CD, 0000-0001-8853-416X

Climate-driven reef decline has prompted the development of next-generation coral conservation strategies, many of which hinge on the movement of adaptive variation across genetic and environmental gradients. This process is limited by our understanding of how genetic and genotypic drivers of coral bleaching will manifest in different environmental conditions. We reciprocally transplanted 10 genotypes of *Acropora cervicornis* across eight sites along a 60 km span of the Florida Reef Tract and documented significant genotype × environment interactions in bleaching response during the severe 2015 bleaching event. Performance relative to site mean was significantly different between genotypes and can be mostly explained by ensemble models of correlations with genetic markers. The high explanatory power was driven by significant enrichment of loci associated DNA repair, cell signalling and apoptosis. No genotypes performed above (or below) bleaching average at all sites, so genomic predictors can provide practitioners with 'confidence intervals' about the chance of success in novel habitats. These data have important implications for assisted gene flow and managed relocation, and their integration with traditional active restoration.

## 1. Introduction

Climate change is rapidly degrading coral reefs, which typically occur near their upper thermal limits. Increases in sea temperature cause coral bleaching, the breakdown of the symbiosis between the coral host and dinoflagellate symbionts in the family *Symbiodiniaceae* [1], resulting in metabolically and physiologically impaired corals. If stress is persistent corals often die [2], compromising the structural and functional integrity of these ecosystems. Coral bleaching has become increasingly frequent [3] and is predicted to impact most of the world's reefs annually by mid-century [4], a recurrent stress on ecosystems that also face local impacts [5,6] and recover slowly even under the best of circumstances [7]. The reduction of greenhouse gas emissions is a requisite for the long-term persistence of coral reef ecosystems, but committed temperature change [8,9] means that oceans will continue to warm for the foreseeable future.

This unprecedented decline has prompted a surge in research on the drivers of coral resilience, highlighting the need for rapid and effective biological interventions that retain ecosystem function [10,11]. These interventions can be broadly categorized as (i) enhancing resilience in the coral holobiont or coral populations, and (ii) repairing damage caused by disturbances [12,13]. Many of these strategies, both proactive and reactive, hinge on actively moving corals and their adaptive variation within and among populations. For example, assisted gene flow, selective breeding, managed relocation and traditional restoration are tools that can alter the genetic composition of coral populations to facilitate adaptive change [12,14].

As the translocation of corals becomes a major component of conservation action, the long-term persistence of coral reefs requires a more developed understanding of the host's genotypic and genetic drivers of thermal tolerance across a

range of environments. Research on broad-sense heritability, genetic and transcriptomic responses to heat stress and genetic–environmental correlations forms the foundation of our understanding of host effects [15], which are conserved across space and time [16–19]. However, these designs are typically limited to one or a few sites and do not examine genotype × environment interactions. A key knowledge gap is how molecular and phenotypic responses to thermal stress will manifest across environmental gradients like those corals will be exposed to in changing climates and as part of conservation and restoration programs.

To address this, we reciprocally transplanted 10 genotypes of the Caribbean coral *Acropora cervicornis* to eight sites and measured bleaching response during the 2015 bleaching event along the Florida Reef Tract. We found significant genotype × environment interactions and significant genotypic effects in a site-adjusted residual bleaching score. These integrative measurements could be predicted with high accuracy from ensemble learning methods of correlated genomic markers, showing that adaptive variance driving bleaching response during a severe warming event is nested within environmental effects. These data are representative of a genotype's performance across many sites under stress and are critical for understanding the implications of next-generation coral conservation.

## 2. Methods

We used next-generation sequencing and phenotypic assays from 668 *Acropora cervicornis* fragments monitored during the 2015 global bleaching event to evaluate genotype × environment interactions and genomic correlates of resilience.

### (a) Reciprocal transplant and bleaching surveys

We collected one genotype of *Acropora cervicornis* from each of 10 sites and propagated them in a common garden nursery for 1 year before outplanting each to 8 of the original collection sites (figure 1). Full details of the outplant experiment can be found in [20]. We created a fully crossed design (with the exception of one genotype at one site, $n = 79$ combinations), with 10 replicate fragments of each genotype at each site. Each replicate was given a visual bleaching score (0–3, electronic supplementary material, figure S1) during May, July and August of 2015, when the global bleaching event significantly impacted the Florida Reef Tract [21]. After removing fragments with early mortality associated with transplantation stress ($n = 122$), we used all remaining fragments ($n = 668$ distributed across all genotypes) to test for a genotype × environment interaction using a two-way ANOVA of the average bleaching score of each fragment from the three time points (after square root transformation). We then calculated the mean bleaching score of all fragments at each site (site mean) and the residual bleaching score of each fragment (individual score − site mean), where high values indicate more bleaching and low values indicate less bleaching. We tested bleaching residuals for a genotype effect using a one-way ANOVA. We measured final survivorship in December 2015.

### (b) Temperature data

Temperatures were logged hourly (onset pendants) at each site from 1 May to 30 September 2015. Data were lost for Jon's Reef. Experiment-wide mean temperatures were above the local bleaching threshold of 30.5°C until at least 15 September. Our temperature data were incomplete after 15 August, so we used the interval where data were available for all sites, which

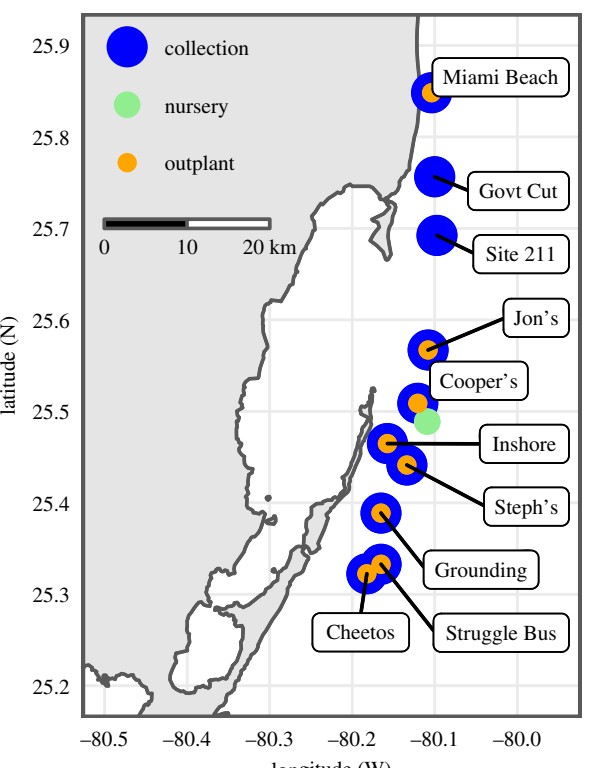

**Figure 1.** Map of collection and outplant sites. *Acropora cervicornis* colonies were collected from each of 10 sites, common gardened in a nursery for 1 year and returned to eight of the original collection sites. Sites span approximately 60 km of the Florida Reef Tract. (Online version in colour.)

includes May 1 to August 15. We calculated summary statistics and used the mean temperature each day as input for principal component analysis from this interval to describe the relationships between sites. We calculated degree heating weeks (DHW) for each site as time spent above 29.7°C, which is one degree above the MMM (August: 28.7°C) at Fowey Rocks (NDBC; FWFY1) in the centre of our study area [22]. Separately, we calculated an experiment-wide DHW metric to summarize the bleaching season using mean temperature of all available sites from 1 May to 30 September, 2015.

### (c) Sequencing data processing

Our sample size ($n = 10$) should capture 80–90% of common alleles (maf > 0.05) in the population [14]. We used sequencing data from [23,24] (sample list available at github.com/druryc/acerv_GxE) to predict number of secondary alleles for corals in the experiment. We aligned demultiplexed reads to the *A. millepora* [25] genome with *bwa mem* [26], called genotype probabilities for filtered loci (quality > 20, mapping quality > 30, present in all 10 samples, SNP *p*-value less than $2 \times 10^{-4}$, per sample depth ≥7) using ANGSD [27]. We exported genotype probabilities for each sample (-doGeno 8, -doPost 1), and summed the probability of the heterozygote (ab) and 2 × secondary homozygote (bb) to predict the number of secondary alleles without hard-calling genotypes. This strategy serves to account for uncertainty in low and variable depth reads while condensing it into a single value suitable for downstream analysis.

We evaluated overall population structure with *PCAngsd* [28] and *NGSadmix* [29] ($k = 1$ to 4) on the genotype probabilities for all sites with minor allele frequency greater than 0.05, determined by minimum log likelihood. We also calculated pairwise identity by state using ANGSD. We aligned all reads to a concatenated symbiont transcriptome [30–33] with *bwa mem* and counted primary alignments with a mapping quality greater than 30 to each genus following [34]. We compared each

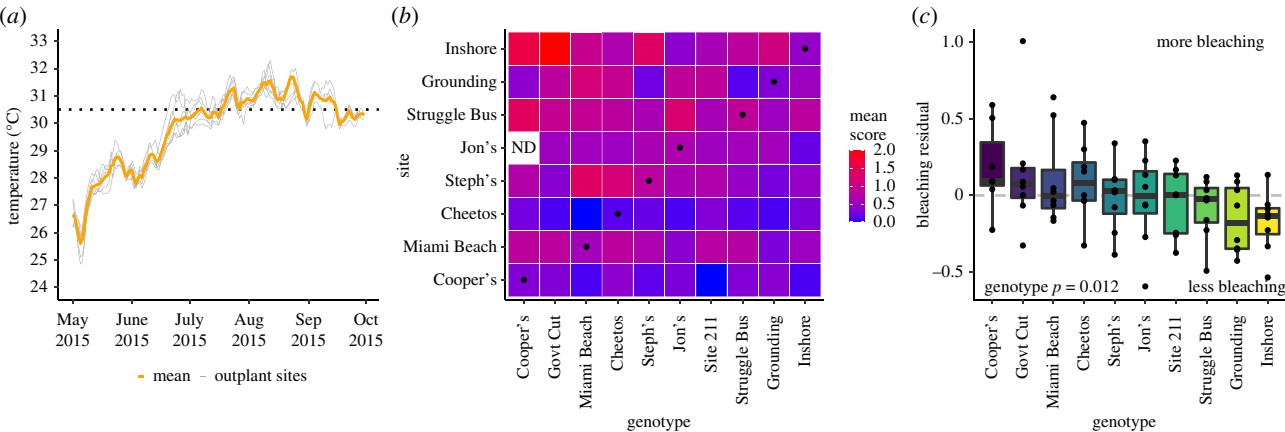

**Figure 2.** Bleaching response. (*a*) Temperature records from seven sites during the 2015 bleaching event, with the overall mean. Dashed line denotes the local bleaching threshold of 30.5°C. (*b*) Mean visual bleaching score over three timepoints (May, June, August) for all genotype × site combinations. Black points denote 'home sites' for eight of 10 genotypes (figure 1). No corals from Jon's genotype were outplanted at Cooper's Reef. (*c*) Site-corrected bleaching residuals (Relative Heat Tolerance) for each genotype. We took the average bleaching score of every fragment and subtracted the site average to correct for environmental differences. Points are visualized as the average at each site for each genotype, showing that no genotypes had absolute performance better or worse than average. Points greater than 0 represent a genotype that underwent more bleaching than average at a given site, while points less than 0 represent less bleaching than average. (Online version in colour.)

genotype's mean bleaching residual against the allelic probability for 13 337 loci passing quality thresholds using linear regression. We removed loci with $p \geq 0.01$ as an initial filtration step, yielding 58 loci with correlation coefficients between 0.73 and 0.89.

## (d) Predicting residuals

We used the 58 loci as input for random forest regression implemented in *caret* [35,36] in R (3.5.0). We used random forests because the ensemble of weak learners is well suited to large numbers of variables and allows us to evaluate additive effects of a 'pre-pruned' dataset of correlated loci using variable importance metrics. To control for overfitting, we used a conservative repeated cross-validation approach ($n = 2$ folds, repeated 20 times) to define the experiment-wide capacity to predict bleaching residuals. We set mtry = 1 to limit the analysis to additive effects (i.e. not interactions) due to our small sample size.

## (e) Annotations and enrichment

We extracted a 1000 bp window (where linkage remains high [25]) from the reference genome centred on each of the 58 loci used for predictions and annotated using blastn against cnidarians (taxid: 6073) with an e-value cutoff of $10^{-5}$. We then extracted a 5000 bp window from all loci ($n = 13\,337$) in the study, aligned them against the Uniprot protein database with blastx and retrieved a gene name and ontology from each sequence. We used these ontologies and the correlation coefficient between allelic likelihood and bleaching residual of all features ($n = 13\,337$) to conduct enrichment analysis using GO_MWU [37].

## (f) Predictions

To create predictions for novel corals not used in the reciprocal transplant, we used additional samples collected as part of ongoing restoration efforts by a network of nurseries along the Florida Reef Tract that had sequencing data [24,38]. We aligned reads and calculated allelic probabilities using the method detailed above for the 58 loci used in the original model.

No additional samples had quality calls at all 58 sites, so we tested the effects of missing data on the original dataset. To do this, we randomly set the number of secondary alleles of varying proportions of variants as missing data, applied the model fit

and evaluated the variation explained by predicted bleaching residuals against bleaching residuals from the original dataset 100 times, a form of internal validation. To apply predictions to new samples, we filtered samples with at least 30 variants (greater than 50% of 58 samples in the original model) and then randomly selected one sample per reef to minimize sampling bias. We used the random forest model presented above on these data to predict relative bleaching tolerance across the Florida Reef Tract. We tested for differences between regions (as defined in [38]) using a one-way ANOVA.

## (g) Validation

To evaluate the power of the random forest model to predict new bleaching phenotypes, we compared predictions to field data for a true validation set using photosynthetic efficiency data from the lower keys (E. Muller 2015, personal communication) during the 2015 bleaching event [39] which also had genomic data [24]. Briefly, photosynthetic efficiency was measured before and after the thermal maximum and we calculated the relative decline (per cent change) for each of the 15 genotypes. After filtering, we compared predicted bleaching residuals from samples with sufficient data ($n = 11$) with these relative declines in fv/fm using linear regression.

## 3. Results

All sites experienced conditions above the bleaching threshold of 30.5°C [40] (figure 2*a*) and 4 of 7 sites experienced at least 40 h above 32°C (table 1). All sites experienced at least 10 DHW (table 1), although this metric was not related to average bleaching by the site (electronic supplementary material, figure S2). We estimate a total of 14 DHW based on mean temperature across all sites. Miami Beach had a distinctive thermal profile, which did not translate to differences in bleaching (electronic supplementary material, figure S2). In the early summer (1 May–15 June), there was approximately 1°C difference in maximum temperature between sites and average temperature and variability were similar (electronic supplementary material, table S1). Importantly, all corals were harboured in a single *in situ* nursery from 1 year prior to outplanting to decrease the influence of acclimatization and isolate host genotypic effects.

**Table 1.** Temperature data for each site (data from Jon's reef was lost) from 1 May to 15 August, logged hourly. Average and s.d. for the duration of the experiment; range is average daily range; max is overall maximum; hours are timepoints logged above each temperature for the duration of the experiment. Temperature stress extended beyond 15 August, but data were incomplete due to instrument failure.

| site | avg | s.d. | range | max | hrs > 30.5℃ | hrs > 31℃ | hrs > 32℃ | hrs > 33℃ | depth (m) | DHW |
|---|---|---|---|---|---|---|---|---|---|---|
| Cooper's | 29.5 | 1.4 | 0.67 | 32.2 | 1112 | 399 | 7 | 0 | 3.4 | 10.6 |
| Cheetos | 29.6 | 1.6 | 1.47 | 33.4 | 1002 | 616 | 82 | 9 | 1.8 | 10.3 |
| Grounding | 29.8 | 1.6 | 1.13 | 32.8 | 1296 | 798 | 88 | 0 | 1.8 | 13.4 |
| Inshore | 29.7 | 1.6 | 0.79 | 32.8 | 1215 | 589 | 40 | 0 | 5.5 | 11.1 |
| Miami Beach | 29.8 | 1.4 | 0.83 | 31.9 | 1433 | 566 | 0 | 0 | 5.5 | 13.7 |
| Steph's | 29.6 | 1.5 | 0.96 | 33.1 | 1132 | 500 | 62 | 1 | 2.1 | 11.5 |
| Struggle Bus | 29.6 | 1.4 | 0.59 | 31.7 | 1280 | 450 | 0 | 0 | 10.7 | 11.7 |

Almost all genotype–site combinations (77 of 79, 97.4%) experienced some visible bleaching attributed to thermal stress (figure 2b, electronic supplementary material, figure S1). There was a significant effect of site ($F = 16.78$, $p < 0.001$) and genotype on bleaching score ($F = 2.87$, $p = 0.002$) and a significant genotype by environment interaction ($F = 1.62$, $p = 0.002$). After calculating residual bleaching score to account for site-based differences, there was a significant effect of genotype (figure 2c; $F = 2.36$, $p = 0.012$). No genotypes were above or below average at every site.

Bleaching score was significantly higher in corals that eventually died than those that recovered from bleaching and survived (December 2015; Wilcox $p = 0.001$; electronic supplementary material, figure S3a). Among all corals in the experiment, 64% of genotype–site combinations experienced 100% mortality, some despite having moderate mean bleaching scores (electronic supplementary material, figure S3b). When examining only corals that did not experience 100% mortality, there was a strong explanatory relationship between bleaching score and subsequent mortality (electronic supplementary material, figure S3c; $R^2 = 0.35$, $p < 0.001$), so this metric is a quality assessment of thermal stress and selective pressure.

Corals in this study were unique genotypes from one population (electronic supplementary material, figure S4). Although the genotype from Cooper's Reef is a genetic outlier in the PCA, NGSadmix identified 1 population, which corresponds to the population structure of *A. cervicornis* from Miami-Dade [24,38]. The 58 loci used in the random forest model were not strongly or significantly correlated with either of the first two principal components driving population structure. One genotype contained a substantial proportion of *Durusdinium*, but all others were dominated (greater than 97% of reads) by *Symbiodinium* (electronic supplementary material, figure S5).

We used random forests to predict bleaching residual using 58 associated genetic markers that were selected based on correlation coefficients from a larger dataset of 13 337 loci. We used twofold repeated cross-validation (randomly choosing five samples, building the model and evaluating on the remaining five, repeated 20 times) to estimate the utility of this method for describing patterns in a 'novel' dataset. Average resampled $R^2$ values were greater than 0.9 (electronic supplementary material, figure S6a; $n = 40$). The best fit captured 96.4% of the variance in

bleaching residuals (electronic supplementary material, figure S6b; $p < 0.001$).

Multiple gene ontologies were significantly enriched (figure 3, FDR < 0.1) in loci with high correlation coefficients including ontologies involving immune responses (GO:0090051), signalling cascades (GO:0032695), apoptosis (GO:0060314), ion transport (GO:009244) and exodeoxyribonucleases (GO:0008852; GO:0034618; GO:003958). Among the 58 loci correlated with bleaching residuals and subsequently used for predictions, 50 had a blastn results against cnidarians. These loci are unlikely to be causative, but are useful for illustrating the utility of machine learning predictions (electronic supplementary material, table S2). The most important variable for predicting bleaching residuals was muscle M-line assembly protein unc-89-like.

We documented $92.1 \pm 3.2\%$ (mean ± 1 s.d.) of variance explained when setting half of the variants to the no information rate (probabilities when no sequencing information was available), indicating that residuals from samples with calls at about half of the 58 loci of interest could still be accurately predicted. We used the same allelic probability strategy presented above for additional samples, calculating expected bleaching residuals for 183 colonies across the Florida Reef Tract. There was a moderate relationship between predicted residuals and photosynthetic decline during bleaching in genotypes from the lower keys ($p = 0.098$, $R^2 = 0.274$) in 11 corals with both types of data (figure 4a).

Predictions yielded a normal distribution of bleaching residuals that was not significantly different from the 10 original genotypes (figure 4b, $p = 0.245$). All regions had samples predicted to be thermally tolerant, but there were no significant differences between regions ($p = 0.339$) and corals predicted to be bleaching tolerant were distributed across the Florida Reef Tract (figure 4c).

## 4. Discussion

Protecting climate resilience in coral reefs is a major goal of contemporary restoration and conservation efforts, but little is known about the interactions between genetic or genotypic drivers and environmental gradients. We show that coral bleaching is governed by substantial genotype × environment interactions and observe large differences in bleaching susceptibility and associated mortality. Underlying this interaction is a genotypic effect, which significantly influences bleaching residuals, a representative site-corrected phenotype. Individual

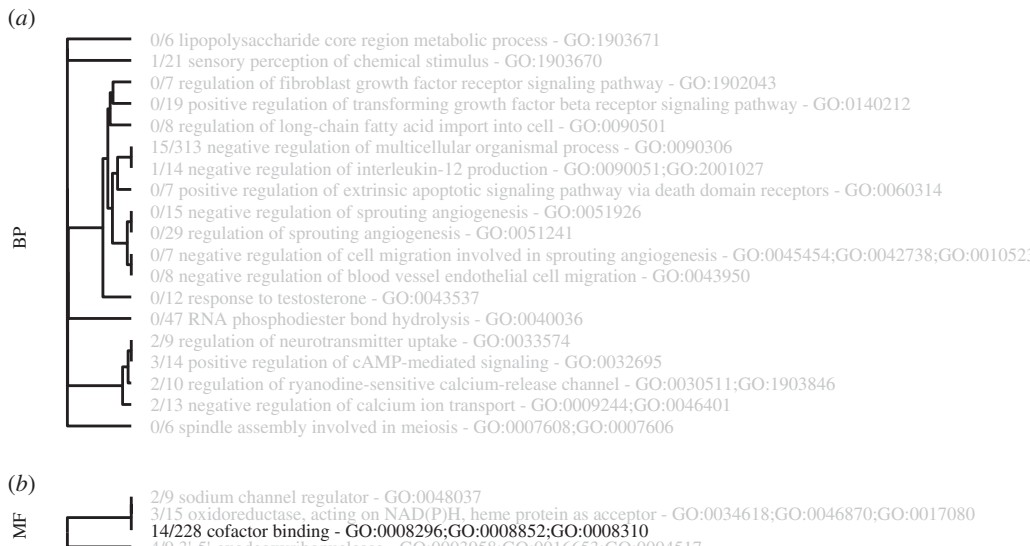

**Figure 3.** Gene ontologies highly correlated with bleaching residuals. Gene ontology enrichments in genes highly correlated with bleaching residuals. Enrichment was calculated from one-way Mann–Whitney U tests on ranked correlation coefficients between bleaching residual for all 13 337 loci using GO_MWU for (*a*) biological processes and (*b*) molecular functions. No cellular compartment ontologies were enriched. Ontologies in black have FDR-adjusted $p < 0.05$ and grey have $p < 0.1$.

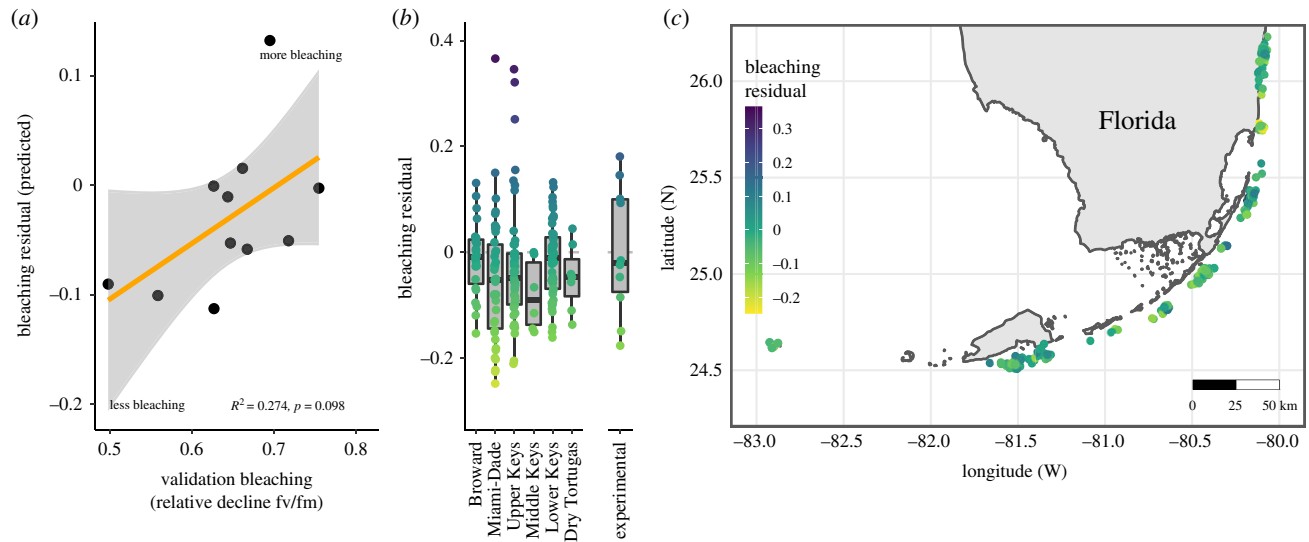

**Figure 4.** Predicted bleaching residuals. We applied the model generated from the reciprocal transplant to genomic data from an additional set of samples with no phenotypic information ($n = 173$). (*a*) Comparison of validation data from the Lower Keys and predicted bleaching residuals. (*b*) Predicted tolerant and susceptible corals were found in every region, with the most resilient individuals found in Miami Beach. (*c*) Map of values, offset to avoid overplotting. (Online version in colour.)

loci associated with genes that are hallmarks of thermal stress in corals were correlated with bleaching residuals, although they are unlikely to be causative given the sparsity of RAD sequencing. These loci were used in ensemble tree-based learning algorithms to predict the relative thermal tolerance of a given genotype with high accuracy. No individuals were bleaching tolerant at all sites. The strongest residuals represent coral colonies that were more (or less) tolerant than their counterparts at greater than 85% (7 of 8) sites, meaning that predictions can account for a portion of the variation associated with interactions and provide practitioners with 'confidence intervals' about their chance of success in a new environment. These tools can be used to understand the importance of the site and genotypic selection and genomic diagnostics for restoration and conservation under climate change, for example, by evaluating phenotypes at a range of sites to understand

likelihoods of achieving performance targets [41] in novel environments or through sequencing studies.

Most reciprocal transplant studies supporting genetic or transcriptomic correlates with thermal stress focus on population×environment changes and/or are conducted at paired sites [42–44], so they are unable to capture robust estimates of genotype × environment interactions. Our results show that genotype × environment interactions should be considered during restoration and conservation because single-site phenotypes are not universal, which has previously been demonstrated in skeletal morphology of corals [45–47]. Genotype × environment interactions reflects phenotypic plasticity. The magnitude of this flexibility about fixed genotypic means can vary and we show that different environments, which may be analogous to changing environments through time, elicit broadly different responses from a given coral.

To the best of our knowledge, only one study has evaluated genotype × environment interactions in coral bleaching [48], finding a highly significant interaction between two sites separated by over 600 km. This study tested the limits of acclimatization by transplanting across a large environmental gradient which exposed corals to seasonal temperature extremes were several degrees outside the norm for home sites. While these distances are plausible or even likely in future restoration and relocation efforts, especially as marine populations move poleward [49], most restoration occurs over finer spatial scales and may need to take advantage of local refugia [50] that are also within the home temperature range of donor populations [51]. Our results support these interactions, but over far more subtle environmental gradients, contradicting the results of [16], which showed high correlations between performance at two sites. Different analytical approaches hinder direct comparisons between these studies, but illustrate the utility of a multi-site evaluation of coral phenotype.

Our data also support the importance of coral genotype in coping with thermal stress, demonstrating that some individuals are naturally more resilient across a variety of sites, but that none are universally above or below average. Only the use of multi-site integrative phenotype like those presented here can resolve this effect, which remains highly relevant as environmental conditions change within and between sites.

The broad dominance of *Symbiodinium* in our samples corresponds to typical patterns in Floridian acroporids [52,53]. This pattern further isolates the influence of the coral animal in our study, although symbiosis also influences host gene expression [54,55], functional variation within symbiont genera is substantial [56] and fine-scale differences between symbiont strains may impact physiology [53]. As expected, many of the canonical stress-response pathways [15] appear to be related to the ability to predict residual bleaching phenotypes in these corals. In particular, our results support cell signalling cascades (GO: 0032695) [57,58], DNA damage/repair mechanisms [59,60] (GO:0008852,GO:0003952) and apoptotic pathways and death-domain receptors that interact with heat shock proteins [61]. The enrichment in multiple ontologies and the high-resolution prediction capacity even with 50% missing data highlight the polygenic nature of heat stress in this system [62].

The 2015 global bleaching event created a mosaic of temperature stress over the area of our experiment, with some sites experiencing nearly 40% more time above the local bleaching threshold than others. Our complete temperature records extend to 15 August, but additional data from a subset of sites showed the experiment-wide average remained above the local bleaching threshold until at least 15 September, so meaningful temperature stress occurred between our final bleaching evaluation (mid-August) and our mortality survey (December). Combined with a time lag and the severe stress of 10–14 DHW, this probably decoupled some of our bleaching scores from the subsequent mortality metrics calculated in December, where 64% of site-genotype combinations experienced 100% morality (see electronic supplementary material, figure S3b,c). For corals that did not undergo complete mortality, there is a high predictive value between bleaching and survivorship, suggesting that this integrative bleaching approach is a robust measure of ecologically relevant stress (and selection pressure) in this system. The high mortality experienced in this study supports the need for distribution of genetic material across environments and the importance of small scale refugia.

Our ability to predict residual bleaching type, a metric which isolates host effects as much as possible, is surprisingly high. While our sample size is limited, the conservative cross-validation approach, integrative multi-site phenotype and anticipated stress-response pathways involved suggest that this is a biologically realistic pattern. Decision tree methods are powerful because they allow many variables to contribute to predictions; when each variable is moderately useful at discerning outcomes, highly accurate models can be created. This strategy has recently been applied in corals for classification problems with multivariate data [63,64] and to gene expression data, showing that stress state is predictable in acroporids [65]. Learning methods may offer substantial advantages for ecological genetics [66] as they have in biomedical research, potentially increasing the predictive power of more traditional methods, such as using polygenic scores and linear models [25]. This strategy also de-emphasizes the importance of individual loci, which may be useful for overcoming the difficulties of developing biomarkers that are highly variable [67]. We only consider our residual prediction to be generalizable to this species in this population, but it serves to illustrate how genetic data can be used to generate 'confidence intervals' about the average likelihood of success in an unknown environment.

For example, our results suggest that there is substantial power in predicting a true validation set of additional genotypes from the Lower Keys that underwent variable amounts of bleaching in the same event as this study. Approximately 27% of the variance in the validation bleaching score (a relative decline of photosynthetic efficiency) could be explained by random forest predicted bleaching residuals, which is a strong signal. This relationship should not be expected to be as strong as the original model fit for logistical and biological reasons. First, our predictions should be integrative of how a sample would perform at a median Miami-Dade site and not necessarily given the conditions the validation corals experienced in the Lower Keys. Second, visual bleaching scores and photosynthetic efficiency can be decoupled. Despite these limitations, this prediction validates the utility of using genomic predictors to assess thermal tolerance and highlights the role of host genomics in the bleaching response.

Exploring natural heat tolerance and leveraging it for selective breeding, restoration and movement of adaptive variation throughout coral populations is scalable and feasible in the near term [13], although it will require considerable effort [68]. To this end, we show that genotype × environment interactions should be addressed within intervention frameworks and that moderately correlated genomic markers can be used to predict an integrative bleaching phenotype, providing practitioners with a 'confidence interval' for success in novel environments. These data provide context for the consideration of site, genotype and genomic diagnostics in coral conservation efforts under climate change.

Ethics. Corals were collected under Biscayne National Park Permits BISC-2014-SCI-0018 and BISC-2015-SCI-0018 and FWC Permit SAL-14-1086-SCRP and SAL-15-1635-SCRP.

Data accessibility. All ecological data, processed sequencing data, a list of specific samples and code associated with this project are available at github.com/druryc/acerv_GxE. Raw sequence data are available at the NCBI Sequence Read Archive under BioProject PRJNA523382 and PRJNA665778.

**Authors' contributions.** C.D. and D.L. conceived the study, collected data, edited the manuscript and approved the final version. C.D. analysed data and wrote the manuscript.

**Competing interest.** We declare we have no competing interests.

**Funding.** C.D. was funded by the Garden Club of America Ecological Restoration Fellowship and MOTE Protect Our Reefs Grant (2013). C.D. acknowledges the Paul G. Allen Family Foundation for support. D.L. acknowledges funding the NOAA Restoration Center (award OAA-NMFS-HCPO-2016–2004840).

**Acknowledgements.** We are grateful to Stephanie Schopmeyer, Dalton Hesley, Kelly Peebles, Michelle Harangody, Evan Morrison, Jay Fisch, Morgan McCall, Ian Enochs, Renee Carlton and Derek Manzello for field and laboratory assistance. We would like to thank Ken Nedimyer, Erich Bartels, Kerry Maxwell, Meaghan Johnson, Liz Goergen and Dave Gilliam for samples from the FRT nursery network. We gratefully acknowledge the technical support and advanced computing resources from University of Hawaii Information Technology Services–Cyberinfrastructure. We appreciate constructive comments on the manuscript from Ariana Huffmyer, Carlo Caruso and Jenna Dilworth.

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
