## [Peer Review File · Proceedings of the Royal Society B: Biological Sciences]

Review History

RSPB-2020-2688.R0 (Original submission)

Review form: Reviewer 1 (Mikhail Matz)

Recommendation

Major revision is needed (please make suggestions in comments)

Scientific importance: Is the manuscript an original and important contribution to its field?

Excellent

General interest: Is the paper of sufficient general interest?

Excellent

Quality of the paper: Is the overall quality of the paper suitable?

Good

Is the length of the paper justified?

Yes

Should the paper be seen by a specialist statistical reviewer?

No

Do you have any concerns about statistical analyses in this paper? If so, please specify them explicitly in your report.

Yes

It is a condition of publication that authors make their supporting data, code and materials available - either as supplementary material or hosted in an external repository. Please rate, if applicable, the supporting data on the following criteria.

Is it accessible?

Yes

Is it clear?

Yes

Is it adequate?

Yes

Do you have any ethical concerns with this paper?

No

Comments to the Author

This is a very well-done and very valuable analysis of genotype x environment interactions in coral bleaching. The experiment is very hard and therefore unique, and the analysis is solid.

That said, I am much less excited about results related to specific loci that seem to be associated with bleaching tolerance - it is my strongest opinion that such analysis cannot be done with just 10 genotypes and sparse GBS data.

So here I go: I am extremely skeptical about the association study (looking for SNPs associated with bleaching phenotype) based on just 10 samples and sparse GBS data. The ability to predict "winner" genotypes in such a setup must surely depend on the overall population structure signal (PCA on Fig S4) of which the indicator SNPs are just correlates, not causes. Listing genes near those SNPs (Fig. 4) is therefore irrelevant - the chance that these specific genes are indeed involved in determining bleaching tolerance is practically zero. Please scrap results and figures involving gene annotations and characterization of individual SNPs.

L119: if using ANGSD, try PCAngsd for pop structure, it is a really powerful and versatile method (uses the beagle formatted data as input, you have that already). Considering that there is a loose separation between winners and losers along the PC1 (Fig, S4), which (I bet a 6-pack) is responsible for the whole cross-validation result with machine learning, it would be cool to identify SNPs strongly associated with PC1 - which is what PCAngsd will spit out almost instantaneously. I bet (the same 6-pack) that those 63 SNPs will be among the strongest PCA signals - which would confirm my intuition about pop structure being responsible for the cross-validation power. Please show me that I am wrong!

Do machine learning predictions for the Florida Reef Tract (Fig 5) match actual bleaching observations? That would be VERY cool (although I would not bet a 6-pack on that, see above...). Importantly, without this information the fact that one can generate predictions of bleaching tolerance does not really prove anything. Of course the prediction algorithm spits out something, but the question is, are those predictions accurate?

Detection of GO signal does suggest that the population structure pattern (Fig S4B) - assuming it drives variation in bleaching - might be driven by selection for some functional differences, which would be exciting indeed! (but still would not justify chasing after individual SNPs and genes, because the inherent sparseness of GBS genotyping practically guarantees that the actual causal loci will be missed). I just want to make sure that the GO_MWU analysis was done correctly, based on all genes that were genotyped (ranked by correlation with bleaching) rather than using

just the 63 top-candidates. I am asking because the Methods text suggests the latter, incorrect, way.

Minor things:

L107: specify what is meant by “common alleles” - I am guessing, derived allele frequency ≥ 0.05 ?

L108: “allelic likelihoods” - this is a rather unusual treatment, people more commonly calculate predicted derived allele count = [prob of heterozygote] + 2[prob of alternative homozygote], as the single-number genotype. It will not be too hard to rerun the regression and random forest analysis with these numbers - I would be very curious to see how the results would change (I expect there will be more power in detecting loci marking “winner” genotypes).

L112: -doGlf 2 (beagle output) does not use the HWE prior when calculating genotype probabilities, it is based exclusively on read counts. To incorporate HWE prior, use -doGlf 8. I would have done that to sharpen the estimates for sites with low read depth, but I leave this up to the authors - there probably will not be much difference since the analysis was restricted to well-covered sites with per-sample depth ≥ 6 , so the effect of the HWE prior would be quite small.

L197: “Loci with high correlation coefficients were significantly enriched (Figure 3, FDR < 0.1) for multiple gene ontologies” - in GO_MWU the logic actually goes the opposite way, so you should say “Multiple GO terms were significantly enriched with genes showing high correlation coefficients.”

Please check the numbers of supplemental figures and references to them in the text, they don't seem to match

Review form: Reviewer 2

Recommendation

Major revision is needed (please make suggestions in comments)

Scientific importance: Is the manuscript an original and important contribution to its field?

Good

General interest: Is the paper of sufficient general interest?

Good

Quality of the paper: Is the overall quality of the paper suitable?

Good

Is the length of the paper justified?

Yes

Should the paper be seen by a specialist statistical reviewer?

No

Do you have any concerns about statistical analyses in this paper? If so, please specify them explicitly in your report.

No

It is a condition of publication that authors make their supporting data, code and materials available - either as supplementary material or hosted in an external repository. Please rate, if applicable, the supporting data on the following criteria.

Is it accessible?

Yes

Is it clear?

Yes

Is it adequate?

No

Do you have any ethical concerns with this paper?

No

Comments to the Author

Drury and Lirman present a manuscript exploring significant genotype by environment interactions in coral during the 2015 natural bleaching event in Florida. The authors used 10 distinct genotypes of *Acropora cervicornis* to create replicate nurseries that were deployed across different reef sites along the Florida Reef Tract. Overall, the manuscript is of high quality and the writing clear, but there are a number of issues that need to be addressed before this manuscript is suitable for publication. Once these issues have been, I would be happy to reconsider this manuscript for publication in Proceedings B.

1. A primary concern is related to a lack of transparency of sequence data used for the analyses. The authors reference a previous manuscript, but provide no other information in the main document or supplementary material. Following the link to the previous paper provided little clarification on quality of sequence data being used. Considering the manuscript appears to be in a short format, it would be worth expanding methods and results section to provide more information about sequencing output, QC, mapping rates etc. It is currently a bit of a black box. I also couldn't find any scripts online to reproduce the analyses. Please make these available on github.
2. Please expand on the methods used for mapping reads to ID symbionts. gg 'combined symbiont reference'. I presume ITS2 haplotypes? Did you test whether the symbiont type was a predictor of bleaching? i.e. did the one housing Clade D fair better than others?
3. It would be helpful to include in the main manuscript more information about how temperature varied across transplant sites. A key assumption is that the local environments at different sites varied during the heatwave, but this is not clear in the main manuscript. At the very least please include Table S1 in main document. Also, do you have any data to show how temperature varied across sites in the build-up to heat stress. Is it possible that certain sites experienced very different temperatures prior to the heat stress that could influence the observed phenotype?
3. The experiment experienced high mortality rates (>64%). It is unclear how many ramets (and genets) were used for the models to predict bleaching outcome and GxE interactions?
4. Is it possible to do a power analysis of some sort to illustrate that the small sample size of this project (n=10) does not impact the results.
5. Were any of the outlier loci under strong linkage? It would be interesting to see if any of these loci are linked or in close proximity on the same chromosome. Also, there is a gene annotation table linked to the *A. millepora* assembly that might be useful. Also please provide blast e-values

for annotations in table S2.

6. Line 206-207: I'm not sure what 'setting half of variants to the no information rate, a loss of about 4% from full data' means

Decision letter (RSPB-2020-2688.R0)

05-Jan-2021

Dear Dr Drury:

I am writing to inform you that your manuscript RSPB-2020-2688 entitled "Genotype by environment interactions in the coral bleaching response" has, in its current form, been rejected for publication in Proceedings B.

This action has been taken on the advice of referees, who have recommended that substantial revisions are necessary. With this in mind we would be happy to consider a resubmission, provided the comments of the referees are fully addressed. However please note that this is not a provisional acceptance.

Sincerely,
Dr Daniel Costa
<mailto:proceedingsb@royalsociety.org>

Associate Editor
Board Member: 1
Comments to Author:
Dear Dr Drury

Your paper has been reviewed by two experts in the field. They agree that your paper has merit and could be published in Proc B provided you are able to address the general and specific comments they have. I trust you will find their comments very constructive even if they might necessitate some rethinking of the analysis and results. I would be happy to consider a revised version of this paper in the new year.

Warm Regards and Season's Greetings, Line K Bay

Reviewer(s)' Comments to Author:

Referee: 1

Comments to the Author(s)

This is a very well-done and very valuable analysis of genotype x environment interactions in coral bleaching. The experiment is very hard and therefore unique, and the analysis is solid.

That said, I am much less excited about results related to specific loci that seem to be associated with bleaching tolerance - it is my strongest opinion that such analysis cannot be done with just 10 genotypes and sparse GBS data.

So here I go: I am extremely skeptical about the association study (looking for SNPs associated with bleaching phenotype) based on just 10 samples and sparse GBS data. The ability to predict "winner" genotypes in such a setup must surely depend on the overall population structure signal (PCA on Fig S4) of which the indicator SNPs are just correlates, not causes. Listing genes near those SNPs (Fig. 4) is therefore irrelevant - the chance that these specific genes are indeed involved in determining bleaching tolerance is practically zero. Please scrap results and figures involving gene annotations and characterization of individual SNPs.

L119: if using ANGSD, try PCAngsd for pop structure, it is a really powerful and versatile method (uses the beagle formatted data as input, you have that already). Considering that there is a loose separation between winners and losers along the PC1 (Fig, S4), which (I bet a 6-pack) is responsible for the whole cross-validation result with machine learning, it would be cool to identify SNPs strongly associated with PC1 - which is what PCAngsd will spit out almost instantaneously. I bet (the same 6-pack) that those 63 SNPs will be among the strongest PCA signals - which would confirm my intuition about pop structure being responsible for the cross-validation power. Please show me that I am wrong!

Do machine learning predictions for the Florida Reef Tract (Fig 5) match actual bleaching observations? That would be VERY cool (although I would not bet a 6-pack on that, see above...). Importantly, without this information the fact that one can generate predictions of bleaching tolerance does not really prove anything. Of course the prediction algorithm spits out something, but the question is, are those predictions accurate?

Detection of GO signal does suggest that the population structure pattern (Fig S4B) - assuming it drives variation in bleaching - might be driven by selection for some functional differences, which would be exciting indeed! (but still would not justify chasing after individual SNPs and genes, because the inherent sparseness of GBS genotyping practically guarantees that the actual causal loci will be missed). I just want to make sure that the GO_MWU analysis was done correctly, based on all genes that were genotyped (ranked by correlation with bleaching) rather than using just the 63 top-candidates. I am asking because the Methods text suggests the latter, incorrect, way.

Minor things:

L107: specify what is meant by "common alleles" - I am guessing, derived allele frequency ≥ 0.05 ?

L108: “allelic likelihoods” - this is a rather unusual treatment, people more commonly calculate predicted derived allele count = [prob of heterozygote] + 2[prob of alternative homozygote], as the single-number genotype. It will not be too hard to rerun the regression and random forest analysis with these numbers - I would be very curious to see how the results would change (I expect there will be more power in detecting loci marking “winner” genotypes).

L112: -doGlf 2 (beagle output) does not use the HWE prior when calculating genotype probabilities, it is based exclusively on read counts. To incorporate HWE prior, use -doGlf 8. I would have done that to sharpen the estimates for sites with low read depth, but I leave this up to the authors - there probably will not be much difference since the analysis was restricted to well-covered sites with per-sample depth ≥ 6 , so the effect of the HWE prior would be quite small.

L197: “Loci with high correlation coefficients were significantly enriched (Figure 3, FDR < 0.1) for multiple gene ontologies” - in GO_MWU the logic actually goes the opposite way, so you should say “Multiple GO terms were significantly enriched with genes showing high correlation coefficients.”

Please check the numbers of supplemental figures and references to them in the text, they don't seem to match

Referee: 2

Comments to the Author(s)

Drury and Lirman present a manuscript exploring significant genotype by environment interactions in coral during the 2015 natural bleaching event in Florida. The authors used 10 distinct genotypes of *Acropora cervicornis* to create replicate nurseries that were deployed across different reef sites along the Florida Reef Tract. Overall, the manuscript is of high quality and the writing clear, but there are a number of issues that need to be addressed before this manuscript is suitable for publication. Once these issues have been, I would be happy to reconsider this manuscript for publication in Proceedings B.

1. A primary concern is related to a lack of transparency of sequence data used for the analyses. The authors reference a previous manuscript, but provide no other information in the main document or supplementary material. Following the link to the previous paper provided little clarification on quality of sequence data being used. Considering the manuscript appears to be in a short format, it would be worth expanding methods and results section to provide more information about sequencing output, QC, mapping rates etc. It is currently a bit of a black box. I also couldn't find any scripts online to reproduce the analyses. Please make these available on github.

2. Please expand on the methods used for mapping reads to ID symbionts. gg ‘combined symbiont reference’. I presume ITS2 haplotypes? Did you test whether the symbiont type was a predictor of bleaching? i.e. did the one housing Clade D fair better than others?

3. It would be helpful to include in the main manuscript more information about how temperature varied across transplant sites. A key assumption is that the local environments at different sites varied during the heatwave, but this is not clear in the main manuscript. At the very least please include Table S1 in main document. Also, do you have any data to show how temperature varied across sites in the build-up to heat stress. Is it possible that certain sites experienced very different temperatures prior to the heat stress that could influence the observed phenotype?

3. The experiment experienced high mortality rates ($>64\%$). It is unclear how many ramets (and genets) were used for the models to predict bleaching outcome and GxE interactions?

4. Is it possible to do a power analysis of some sort to illustrate that the small sample size of this project (n=10) does not impact the results.
5. Were any of the outlier loci under strong linkage? It would be interesting to see if any of these loci are linked or in close proximity on the same chromosome. Also, there is a gene annotation table linked to the *A. millepora* assembly that might be useful. Also please provide blast e-values for annotations in table S2.
6. Line 206-207: I'm not sure what 'setting half of variants to the no information rate, a loss of about 4% from full data' means

Author's Response to Decision Letter for (RSPB-2020-2688.R0)

See Appendix A.

RSPB-2021-0177.R0

Review form: Reviewer 1 (Mikhail Matz)

Recommendation

Accept as is

Scientific importance: Is the manuscript an original and important contribution to its field?

Excellent

General interest: Is the paper of sufficient general interest?

Excellent

Quality of the paper: Is the overall quality of the paper suitable?

Excellent

Is the length of the paper justified?

Yes

Should the paper be seen by a specialist statistical reviewer?

No

Do you have any concerns about statistical analyses in this paper? If so, please specify them explicitly in your report.

No

It is a condition of publication that authors make their supporting data, code and materials available - either as supplementary material or hosted in an external repository. Please rate, if applicable, the supporting data on the following criteria.

Is it accessible?

Yes

Is it clear?

Yes

Is it adequate?

Yes

Do you have any ethical concerns with this paper?

Yes

Comments to the Author

I am fully satisfied with the response, which was a pleasure to read. While I am still skeptical about the surprising accuracy of genome-based bleaching predictions, I do owe you a 6-pack for demonstrating that my proposed explanation was wrong, and at the moment I cannot think of any other way how the result can be an artifact. Let the history be the final judge!

Review form: Reviewer 2

Recommendation

Accept with minor revision (please list in comments)

Scientific importance: Is the manuscript an original and important contribution to its field?

Good

General interest: Is the paper of sufficient general interest?

Good

Quality of the paper: Is the overall quality of the paper suitable?

Good

Is the length of the paper justified?

Yes

Should the paper be seen by a specialist statistical reviewer?

No

Do you have any concerns about statistical analyses in this paper? If so, please specify them explicitly in your report.

No

It is a condition of publication that authors make their supporting data, code and materials available - either as supplementary material or hosted in an external repository. Please rate, if applicable, the supporting data on the following criteria.

Is it accessible?

Yes

Is it clear?

Yes

Is it adequate?

Yes

Do you have any ethical concerns with this paper?

No

Comments to the Author

The authors have done well to address all of the concerns raised in the previous round of reviews. Nice work. A few small typos throughout that need to be addressed.

Decision letter (RSPB-2021-0177.R0)

08-Feb-2021

Dear Dr Drury

I am pleased to inform you that your manuscript RSPB-2021-0177 entitled "Genotype by environment interactions in the coral bleaching response" has been accepted for publication in Proceedings B.

The referee(s) have recommended publication, but also suggest some minor revisions to your manuscript. Therefore, I invite you to respond to the referee(s)' comments and revise your manuscript. Because the schedule for publication is very tight, it is a condition of publication that you submit the revised version of your manuscript within 7 days. If you do not think you will be able to meet this date please let us know.

Online supplementary material will also carry the title and description provided during submission, so please ensure these are accurate and informative. Note that the Royal Society will not edit or typeset supplementary material and it will be hosted as provided. Please ensure that

the supplementary material includes the paper details (authors, title, journal name, article DOI). Your article DOI will be 10.1098/rspb.[paper ID in form xxxx.xxxx e.g. 10.1098/rspb.2016.0049].

Sincerely,
Dr Daniel Costa
mailto:proceedingsb@royalsociety.org

Associate Editor
Board Member
Comments to Author:
Dear Dr Drury

I am pleased that the original reviewers are happy with your responses to their concerns and I echo this sentiment. A few typos remain so please have a final look at the text.

Warm Regards, Line K Bay

Reviewer(s)' Comments to Author:

Referee: 2

Comments to the Author(s).

The authors have done well to address all of the concerns raised in the previous round of reviews. Nice work. A few small typos throughout that need to be addressed.

Luke Thomas

Referee: 1

Comments to the Author(s).

I am fully satisfied with the response, which was a pleasure to read. While I am still skeptical about the surprising accuracy of genome-based bleaching predictions, I do owe you a 6-pack for demonstrating that my proposed explanation was wrong, and at the moment I cannot think of any other way how the result can be an artifact. Let the history be the final judge!

Author's Response to Decision Letter for (RSPB-2021-0177.R0)

See Appendix B.

Decision letter (RSPB-2021-0177.R1)

09-Feb-2021

Dear Dr Drury

I am pleased to inform you that your manuscript entitled "Genotype by environment interactions in the coral bleaching response" has been accepted for publication in Proceedings B.

Your article has been estimated as being 9 pages long. Our Production Office will be able to confirm the exact length at proof stage.

Open Access

You are invited to opt for Open Access, making your freely available to all as soon as it is ready for publication under a CCBY licence. Our article processing charge for Open Access is £1700. Corresponding authors from member institutions

Paper charges

Sincerely,

Appendix A

Comments to the Author(s)

This is a very well-done and very valuable analysis of genotype x environment interactions in coral bleaching. The experiment is very hard and therefore unique, and the analysis is solid.

That said, I am much less excited about results related to specific loci that seem to be associated with bleaching tolerance - it is my strongest opinion that such analysis cannot be done with just 10 genotypes and sparse GBS data.

So here I go: I am extremely skeptical about the association study (looking for SNPs associated with bleaching phenotype) based on just 10 samples and sparse GBS data. The ability to predict "winner" genotypes in such a setup must surely depend on the overall population structure signal (PCA on Fig S4) of which the indicator SNPs are just correlates, not causes. Listing genes near those SNPs (Fig. 4) is therefore irrelevant - the chance that these specific genes are indeed involved in determining bleaching tolerance is practically zero. Please scrap results and figures involving gene annotations and characterization of individual SNPs.

L119: if using ANGSD, try PCAngsd for pop structure, it is a really powerful and versatile method (uses the beagle formatted data as input, you have that already). Considering that there is a loose separation between winners and losers along the PC1 (Fig, S4), which (I bet a 6-pack) is responsible for the whole cross-validation result with machine learning, it would be cool to identify SNPs strongly associated with PC1 - which is what PCAngsd will spit out almost instantaneously. I bet (the same 6-pack) that those 63 SNPs will be among the strongest PCA signals - which would confirm my intuition about pop structure being responsible for the cross-validation power. Please show me that I am wrong!

We re-ran `angsd` following the suggestions below (`-doGeno 8` with `af prior`, predicted derived allele count), yielding 58 loci with large correlations ($p < 0.01$, $r \sim 0.7$) between allele count and bleaching residuals. We then compared each of these 58 to the new PC1 loadings (PCAngsd) to examine if they are signals/correlated with population structure. The highest absolute correlation of these values was (-0.12) , suggesting that they are not associated with PC1. None of the comparisons had a significant p-value (smallest observed p-value for the regression is 0.26). We also point out that in the original (and revised) PCA that there is actually NOT a strong relationship ($r^2 = 0.0001, p = 0.97$ along PC1 with bleaching residual: high and low values cluster together (left, yellow and purple).

Because of these results and our new validation, we have elected to keep information about specific loci in the manuscript, although we have moved these data to a supplemental table, removed the figure and now emphasize that they should not be considered causative for the reasons mentioned above. Nevertheless, we believe they should be a resource available as part of the manuscript and have not totally removed them.

We will let you, mystery reviewer, decide what this means about beer.

Do machine learning predictions for the Florida Reef Tract (Fig 5) match actual bleaching observations? That would be VERY cool (although I would not bet a 6-pack on that, see above...). Importantly, without this information the fact that one can generate predictions of bleaching tolerance does not really prove anything. Of course the prediction algorithm spits out something, but the question is, are those predictions accurate?

After confirming that the SNPs of interest are not primarily predicting population structure, we re-ran the same random forest analysis as previously described and observed a small increase in accuracy (95.9% R^2 in 20x 2-fold cross validation).

We were able to find some validation data (after extensive searching during the original writing of the manuscript) courtesy of Erinn Muller at MOTE in the lower keys. In total, 15 genotypes had before-after fv/fm data during the 2015 bleaching event in the keys, 11 of which had genomic data in our additional predicted samples. We have added this analysis to the manuscript and show that there is a marginally insignificant relationship between these two metrics that explains about 27% of variance. We think this is pretty cool and appreciate the suggestion! There is now some additional discussion about this validation and how/why it may be a poorer match than the internal validation set, but we hope the reviewer agrees that this is a meaningful step forward.

Given the success of this additional analysis, we have elected to keep the prediction data in the manuscript so it will be available to practitioners.

Detection of GO signal does suggest that the population structure pattern (Fig S4B) - assuming it drives variation in bleaching - might be driven by selection for some functional differences, which would be exciting indeed! (but still would not justify chasing after individual SNPs and genes, because the inherent sparseness of GBS genotyping practically guarantees that the actual causal loci will be missed). I just want to make sure that the GO_MWU analysis was done correctly, based on all genes that were genotyped (ranked by correlation with bleaching) rather than using just the 63 top-candidates. I am asking because the Methods text suggests the latter, incorrect, way.

We appreciate the reviewer pointing out this ambiguity and have clarified the text, and note that we did conduct this analysis correctly. We conducted GO_MWU analysis on all 13k loci in the study, using the correlation coefficient between the newly recalculated derived allele count and the bleaching residual as the 'heats'. The text has been adjusted to reflect this.

Minor things:

L107: specify what is meant by “common alleles” - I am guessing, derived allele frequency ≥ 0.05 ?

This is correct and has been amended in the text.

L108: “allelic likelihoods” - this is a rather unusual treatment, people more commonly calculate predicted derived allele count = [prob of heterozygote] + 2[prob of alternative homozygote], as the single-number genotype. It will not be too hard to rerun the regression and random forest analysis with these numbers - I would be very curious to see how the results would change (I expect there will be more power in detecting loci marking “winner” genotypes).

L112: -doGlf 2 (beagle output) does not use the HWE prior when calculating genotype probabilities, it is based exclusively on read counts. To incorporate HWE prior, use -doGlf 8. I would have done that to sharpen the estimates for sites with low read depth, but I leave this up to the authors - there probably will not be much difference since the analysis was restricted to well-covered sites with per-sample depth ≥ 6 , so the effect of the HWE prior would be quite small.

We have made an effort to sort out this comment and think it is actually meant to be -doGeno 8, which calculates posterior probabilities of all three genotypes and can use a pop-wide allele frequency as a prior. -doGlf controls file dumping.

Assuming this is correct, we have recalculated genotype *probabilities* and merged this with the previous comment on allelic likelihoods. We now calculate predicted derived allele count as mentioned above and use this value for regressions and random forests. We appreciate this suggestion.

L197: “Loci with high correlation coefficients were significantly enriched (Figure 3, FDR < 0.1) for multiple gene ontologies” - in GO_MWU the logic actually goes the opposite way, so you should say “Multiple GO terms were significantly enriched with genes showing high correlation coefficients.”

Ah, we greatly appreciate this clarification and have adjusted the text as such.

Please check the numbers of supplemental figures and references to them in the text, they don't seem to match

We appreciate the reviewer noticing this and believe the figures and text now match.

Referee: 2

Drury and Lirman present a manuscript exploring significant genotype by environment interactions in coral during the 2015 natural bleaching event in Florida. The authors used 10 distinct genotypes of *Acropora cervicornis* to create replicate nurseries that were deployed across different reef sites along the Florida Reef Tract. Overall, the manuscript is of high quality and the writing clear, but there are a number of issues that need to be addressed before this manuscript is suitable for publication. Once these issues have been, I would be happy to reconsider this manuscript for publication in Proceedings B.

1. A primary concern is related to a lack of transparency of sequence data used for the analyses. The authors reference a previous manuscript, but provide no other information in the main document or supplementary material. Following the link to the previous paper provided little clarification on quality of sequence data being used. Considering the manuscript appears to be in a short format, it would be worth expanding methods and results section to provide more information about sequencing output, QC, mapping rates etc. It is currently a bit of a black box. I also couldn't find any scripts online to reproduce the analyses. Please make these available on github.

We apologize for the confusion. All cluster/bash scripts were available on github at the time of initial submission at the link in the data availability section, but could be found as commented lines within the R script. We originally uploaded this code this way to preserve the sequence of our analysis (which was primarily conducted in R), but have separated it into bash/R scripts at github.com/druryc/acerv_GxE. We have also uploaded lists of the individual files within the previously documented BioProjects for complete reproducibility of this study.

2. Please expand on the methods used for mapping reads to ID symbionts. gg 'combined symbiont reference'. I presume ITS2 haplotypes? Did you test whether the symbiont type was a predictor of bleaching? i.e. did the one housing Clade D fair better than others?

This technique has been used in several manuscripts to date (2-4) and is based on a concatenated reference of 4 transcriptomes (not single genes/ITS2) from *Symbiodinium*, *Breviolum*, *Cladocopium* and *Durusdinium*. While evaluating this comment we found a new version of one transcriptome had been uploaded, so we re-ran the analysis, the only basic change was the elimination of small numbers of alignments to *Breviolum*, which were insignificant to begin with.

We have not substantially adjusted the text, but have amended the following line

- "We aligned all reads to a concatenated symbiont reference with *bwa mem* (5-8) and counted primary alignments with a mapping quality >30 to each genera following (3)."
- and provided the links/manuscripts for all of these data in the references
- and added additional information for reproductibility in the code for this analysis (and the references/links) can also be found at github.com/druryc/acerv_GxE.

We did not test if symbiont genotype is a predictor of bleaching because only a single genet has a meaningful proportion of anything besides *Symbiodinium*, 'Govt Cut'. This pattern is expected, as most Floridian *A. cervicornis* harbors primarily *Symbiodinium*. We note that this

genotype had the 2nd worst performance (bleaching residual, F2C) and that this is a relatively 'inshore' population, which has been shown to harbor *Durusdinium* in previous work (9).

We feel that discussion of this in the manuscript would be overstepping our ability to make conclusions and therefore elect to leave it out.

3. It would be helpful to include in the main manuscript more information about how temperature varied across transplant sites. A key assumption is that the local environments at different sites varied during the heatwave, but this is not clear in the main manuscript. At the very least please include Table S1 in main document. Also, do you have any data to show how temperature varied across sites in the build-up to heat stress. Is it possible that certain sites experienced very different temperatures prior to the heat stress that could influence the observed phenotype?

We appreciate this comment have moved the table to the main text. We do not have any additional data beyond the setup of this experiment and since these were sites established for the purpose of this project and not typically used by other stakeholders (scientists, restoration groups, etc.) there is no high resolution (spatial or temporal) temperature data available outside of this experiment.

To clarify, the corals in this experiment were housed in a common garden for 1 year prior to outplanting (10) with the explicit goal of standardizing long-term/preceding temperature dynamics to isolate genotype effects. The temperature records in the manuscript reflect the complete time-series (i.e. started at the time of outplanting) for the corals involved.

To address the question about temperature patterns in the build-up to stress, we included additional information on mean and variability in the 'early stages' before the temperature crossed the bleaching threshold (May 1 to June 15) as the new ST1, but have not been able to add earlier data. We also include an overall summary of temperature patterns in SF2. If the reviewer has specific calculations about the early temperatures we are happy to include them, but note that the patterns in F1a are pretty consistent across sites.

3. The experiment experienced high mortality rates (>64%). It is unclear how many ramets (and genets) were used for the models to predict bleaching outcome and GxE interactions?

We are unsure about the reviewer's exact question here, but have tried to clarify the methods. Bleaching was evaluated in May, July and August. Some corals experienced transplantation related stress before May (n=112) and were not included in this analysis, yielding 668 remaining fragments of 10 genotypes. We calculated the mean score for all three timepoints as the individual bleaching score for each fragment to produce a holistic bleaching severity metric that was not as sensitive to the timepoint collected. For the GxE two-way anova, we used this data from 668 ramets total, coming from all ten genotypes after having removed 112 ramets (from 790 original) that experienced transplantation related mortality (before temperature stress). We then compared differences between genotypes of the bleaching residual (not raw mean bleaching score) using the same 668 fragments. Recovery (or mortality) surveys were conducted in December, which is where the 64% figure comes from. We hope this clarifies the reviewer's question.

- "After removing fragments with early mortality associated with transplantation stress (n=112), we used all remaining fragments (n=668 of 10 genotypes) to test

for a genotype x environment interaction using a two-way ANOVA after square root transformation of the average bleaching score of each fragment at the three time points.”

- “We tested bleaching residuals for a genotype effect using a one-way ANOVA. We measured final survivorship in December 2015.”

4. Is it possible to do a power analysis of some sort to illustrate that the small sample size of this project (n=10) does not impact the results.

We have made a good faith effort to figure out how to address this and do not know how we can. We acknowledge that our small (genotypic) sample size is a limitation, but point out that previous work suggests this should capture most of the common (maf>0.05) genetic diversity in the population (11).

Normally, the small sample size would be more concerning if we were unable to document the pattern due to lack of power, but in this case we resolved significant differences, so we did have enough power. Since our corals and sites were arbitrarily chosen along the FRT without any *a priori* information about bleaching susceptibility or thermal patterns at any site, they should be an appropriate random sample.

5. Were any of the outlier loci under strong linkage? It would be interesting to see if any of these loci are linked or in close proximity on the same chromosome. Also, there is a gene annotation table linked to the *A. millepora* assembly that might be useful. Also please provide blast e-values for annotations in table S2.

We have included the e-values in the table now and note that the annotations from *A. millepora* are included in our blast results (and was our reference). We examined linkage before the original submission but did not find any interesting patterns. After re-analyzing the data we have re-examined this, but there has been no change. There is a region on Chromosome 3 where two SNPs separated by 467bp have a high correlation (R=0.81), but we were unable to annotate these loci (chr3_15009763 and chr3_15010230 in ST2). No other loci that are correlated with bleaching and with high linkage values are within 0.75MB given the sparsity of our data. Given R1s useful commentary about deprioritizing the influence of individual loci we do not feel that this is very strong evidence and have elected not to include this in the manuscript.

6. Line 206-207: I’m not sure what ‘setting half of variants to the no information rate, a loss of about 4% from full data’ means.

We have clarified this in the text, it refers to our predicted samples that did not have sequencing information at all 58 loci used in the original prediction. We tested how much data could be removed and still retrieve accurate predictions. “We documented $92.1 \pm 3.2\%$ (mean $\pm 1sd$) of variance explained when setting half of variants to the no information rate (genotype probabilities when no sequencing information was available) , indicating that samples with calls at only half of the 58 loci of interest could still be accurately predicted.”

References

1. Z. L. Fuller *et al.*, Population genetics of the coral *Acropora millepora*: Toward genomic prediction of bleaching. *Science* **369**, (2020).
2. C. Drury, R. Pérez Portela, X. M. Serrano, M. Oleksiak, A. C. Baker, Fine-scale structure among mesophotic populations of the great star coral *Montastraea cavernosa* revealed by SNP genotyping. *Ecology and Evolution*, (2020).
3. D. P. Manzello *et al.*, Role of host genetics and heat tolerant algal symbionts in sustaining populations of the endangered coral *Orbicella faveolata* in the Florida Keys with ocean warming. *Global Change Biology* **25**, 1016-1031 (2019).
4. A. B. Sturm, R. J. Eckert, J. G. Méndez, P. González-Díaz, J. D. Voss, Population genetic structure of the great star coral, *Montastraea cavernosa*, across the Cuban archipelago with comparisons between microsatellite and SNP markers. *Scientific reports* **10**, 1-15 (2020).
5. T. Bayer *et al.*, Symbiodinium transcriptomes: genome insights into the dinoflagellate symbionts of reef-building corals. *PLoS One* **7**, e35269 (2012).
6. A. J. Bellantuono, K. E. Dougan, C. Granados-Cifuentes, M. Rodriguez-Lanetty, Free-living and symbiotic lifestyles of a thermotolerant coral endosymbiont display profoundly distinct transcriptomes under both stable and heat stress conditions. *Molecular Ecology* **28**, 5265-5281 (2019).
7. S. W. Davies, J. B. Ries, A. Marchetti, K. D. Castillo, Symbiodinium functional diversity in the coral *Siderastrea siderea* is influenced by thermal stress and reef environment, but not ocean acidification. *Frontiers in Marine Science* **5**, 150 (2018).
8. S. Davies.
9. I. B. Baums, M. E. Johnson, M. K. Devlin-Durante, M. W. Miller, Host population genetic structure and zooxanthellae diversity of two reef-building coral species along the Florida Reef Tract and wider Caribbean. *Coral Reefs* **29**, 835-842 (2010).
10. C. Drury, D. Manzello, D. Lirman, Genotype and local environment dynamically influence growth, disturbance response and survivorship in the threatened coral, *Acropora cervicornis*. *PLoS One* **12**, e0174000 (2017).
11. I. B. Baums *et al.*, Considerations for maximizing the adaptive potential of restored coral populations in the western Atlantic. *Ecological Applications* **29**, e01978 (2019).

Appendix B

I am pleased that the original reviewers are happy with your responses to their concerns and I echo this sentiment. A few typos remain so please have a final look at the text.

Warm Regards, Line K Bay

Reviewer(s)' Comments to Author:

Referee: 1

Comments to the Author(s).

I am fully satisfied with the response, which was a pleasure to read. While I am still skeptical about the surprising accuracy of genome-based bleaching predictions, I do owe you a 6-pack for demonstrating that my proposed explanation was wrong, and at the moment I cannot think of any other way how the result can be an artifact. Let the history be the final judge!

Referee: 2

Comments to the Author(s).

The authors have done well to address all of the concerns raised in the previous round of reviews. Nice work. A few small typos throughout that need to be addressed.

Luke Thomas

We appreciate both reviewer's positive feedback and constructive suggestions about the initial version of the manuscript. We have made small adjustments to the text and corrected typos.